# Pain as Defining Feature of Health Status and Prominent Therapeutic Target in Patients with Hidradenitis Suppurativa

**DOI:** 10.3390/jcm10163648

**Published:** 2021-08-18

**Authors:** Francesca Sampogna, Irene Campana, Luca Fania, Simona Mastroeni, Roberta Fusari, Davide Ciccone, Sabatino Pallotta, Damiano Abeni

**Affiliations:** 1Clinical Epidemiology Unit, IDI-IRCCS, 00167 Rome, Italy; s.mastroeni@idi.it (S.M.); r.fusari@idi.it (R.F.); d.abeni@idi.it (D.A.); 2Dermatology Department, IDI-IRCCS, 00167 Rome, Italy; irenecamp89@gmail.com (I.C.); l.fania@idi.it (L.F.); d.ciccone@idi.it (D.C.); s.pallotta@idi.it (S.P.)

**Keywords:** clinical severity, health status, hidradenitis suppurativa, pain, therapy

## Abstract

Background. Pain is one of the main aspects of hidradenitis suppurativa that strongly affects the quality of life of patients. We explored the relationship between pain and clinical severity as well as its role in defining the health status in patients with HS. Methods. Pain was defined by three measures: (a) question 1 (“my skin hurts”) of the Skindex-17; (b) Bodily Pain (BP) scale of the SF-36; and (c) Visual Analog Scale (VAS). Clinical severity of HS was assessed by the Hurley staging, the Sartorius HS Score, and the International HS Severity Score System. Results. The study population included 341 HS patients with complete data for the VAS pain, 316 for question 1 of the Skindex-17, and 294 for BP. Clinical severity was positively associated with pain. This result was observed for all three severity scores and all three pain evaluation methods. In addition, the number of fistulae, abscesses, and nodules were significantly associated with the three severity measures of pain, while the association with scars was not observed for question 1 of the Skindex-17 and BP. Conclusions. Pain may be a good proxy of clinical severity and efficacy of a treatment in HS and therefore a crucial hallmark of patients’ health status.

## 1. Introduction

Hidradenitis suppurativa/acne inversa (HS) is a chronic, inflammatory skin condition characterized by abscesses in the apocrine gland-bearing areas of the body [1]. Data on the prevalence of HS are not concordant, with an estimation from less than 0.1% [2] up to 4% [3] or more. Both genetic and environmental factors contribute to the disease [4]. The diagnosis is based on the clinical characteristics of the skin lesions; the localization in the predisposed areas, usually in the anogenital regions, axillary, and inframammary folds and groin; and the chronicity and recurrence of the cutaneous lesions [5].

Due to recurrent flares, sinus tracts, abscesses, fistulas, and scarring, HS is an extremely painful condition [6]. The 1994 International Association for the Study of Pain stated that “pain is an unpleasant sensory and emotional experience associated with actual or potential tissue damage or described in terms of such damage” [7]. Patients with HS may experience intense pain. It has been described as hot, burning, stretching, pressure, sharp, and splitting [8]. Patients consider it as the most important contributor to their quality of life impairment [8,9,10] and as one of the most intolerable features of the disease [11]. In addition to smell, suppuration, and pruritus, pain severely affects physical and mental health [12].

A large European case-control multicenter study showed pain to be the highest in patients with HS among several other dermatological conditions [13,14]. Additionally, HS patients reported worse bodily pain and mental health compared to several non-dermatological chronic diseases [15]. Recently, pain has been recognized as one of the core outcomes for the design of HS clinical trials [16]. Studies on pain in HS are mainly focused on its management in clinical practice [17,18]. Other studies have investigated the specific qualitative aspects of pain [6] and its relationship with quality of life and psychiatric comorbidities [8,19,20].

It is reasonable to assume that pain may be strongly associated with clinical severity as measured by the classical severity indexes, such as Hurley staging, Sartorius score, or IHS4.

Our hypothesis was that pain may be a good proxy for the evaluation of clinical severity in HS and that even a simple measurement, such as a VAS scale, may provide useful, comprehensive information.

The purpose of this study was thus to explore the relationship between pain and clinical severity as well as its role in defining the state of health in HS patients by using different pain measures.

## 2. Materials and Methods

### 2.1. Study Design

This is a cross-sectional, observational study on patients with HS conducted in a research hospital. The study has been approved by the Institutional Ethical Committee of IDI-IRCCS, Rome, Italy (protocol number 608-1).

### 2.2. HS Patients

Consecutive patients with HS were recruited between August 2016 and October 2019 at the dedicated HS outpatient clinic of IDI-IRCSS, a national dermatological reference center in Rome, Italy.

Patients aged 16 years or more; with a new diagnosis of HS or presenting for the first time to the hospital; with a history of at least 6 months of nodules, abscesses, fistulae, and secondary retracting scars; affected intertriginous sites; and who signed an informed consent were included in the study.

Patients unable to understand the Italian language or the questions or those with major psychiatric conditions were excluded.

### 2.3. Measures of Pain

1.Question 1 of the Skindex-17

The Skindex-17 [21] measures dermatology-related quality of life on two subscales (symptoms and psychosocial). The first item of the instrument (“My skin hurts”) asks about the frequency of pain due to the skin condition during the previous 4 weeks, with possible answers ‘‘never’’ (score = 0), ‘‘rarely/sometimes’ (1), and ‘‘often/always’ (2).

2.Bodily Pain (BP) scale of the SF-36

Pain was also evaluated using the BP scale of the SF-36, which is a generic indicator of health status [22]. The BP scale is one of the eight scales composing the SF-36. It has two items that measure body pain intensity and the extent to which pain interferes with daily activities. Scores range from 0 to 100, with higher scores indicating a better status.

3.Visual Analog Scale (VAS) of pain

Pain during the previous week was also evaluated using a VAS from 0 (no pain) to 10 (worst imaginable pain).

### 2.4. Clinical Severity Measures

Clinical severity of HS was assessed by the Hurley staging [23], the Sartorius HS Score [24], and the International HS Severity Score System (IHS4) [25]. The Hurley system describes three clinical severity stages. The Sartorius HS Score is based on the anatomic region involved, the number of abscesses, nodules, fistulas, scars, and distance between lesions. The IHS4 is the result of a formula including the number of nodules, abscesses, and draining tunnels (fistulae, sinuses). HS is defined as mild/moderate if the score is <11 points and severe if the score is ≥11 points.

### 2.5. Statistical Analysis

Mean scores of the three clinical severity measures (Hurley staging, Sartorius HS Score, and IHS4) were compared in different levels of pain scores using one-way ANOVA.

Question 1 of the Skindex-17 was analyzed according to the original scoring (never, rarely/sometimes, often/all the time). BP was categorized as <25, 25–49, and 50 or higher. VAS pain was categorized as <5, 5–6, or ≥7. Additionally, the different components of severity, i.e., mean number of fistulae, abscesses, nodules, and scars, were compared according to the level of pain.

Tukey’s post-hoc test was used for pairwise comparisons. Mean SF-36 scale scores were graphically represented according to the different levels of pain. For question 1 of the Skindex-17, two groups were created: one experiencing pain never/rarely/sometimes and the other often/always. Mean values were compared using Mann-Whitney U test.

All analyses were performed using IBM SPSS Statistics for Windows, Release 26.0.0.1 (IBM Corp., Armonk, NY, USA).

## 3. Results

The study population included 341 HS patients with complete data for the VAS pain, 316 for question 1 of the Skindex-17, and 294 for the BP scale of the SF-36. There were 131 males (38.4%) in the first group, 127 (40.2%) in the second group, and 111 (37.8%) in the third group. Mean age (standard deviation) was 32.4 (11.9), 33.3 (11.6), and 32.6 (11.7) years, respectively.

Severity was significantly, positively associated with pain (Table 1).

This result was observed for all severity and pain measures. As to the different components of clinical severity, the number of fistulae, abscesses, and nodules were significantly associated with the three severity measures of pain. On the contrary, the association with scars was not observed for two out of the three pain scales (i.e., Skindex-17 question 1 and BP). The comparison of subgroups by pairs showed that, in general, differences were significant between the group with the highest level of pain and each of the other groups.

Mean values of the 8 scales of the SF-36 are graphically represented in Figure 1 in the three categories of VAS pain. Differences between the extreme categories (i.e., <5 and 7 or higher) were all statistically significant. The difference was significant between categories <5 and 5–6.99 for PF, BP, SF, and RE and for the comparison “5–6.99” vs. “≥7” for PF, RP, and BP.

Figure 2 represents mean values of the SF-36 subscales for the groups of patients answering “never/rarely/sometimes” vs. “often always” to question 1 of the Skindex-17. All scale scores were significantly lower in patients answering “often/always” than in the other group.

In Figure 3, the mean values of the 8 scale scores are represented for three categories of the BP scale. The scores were significantly different for all the subscales when comparing all the categories among them except for the scales GH, VT, and MH between the scores 25–49 and 50 or higher.

## 4. Discussion

In this study, we reported a very strong association between pain and clinical severity measures in patients with HS. This result was observed for all three severity scores and all three pain evaluation methods we used. The issue of pain in skin diseases is noteworthy, and its diagnosis and treatment are an important aspect in the management of the diseases. HS-related pain may be acute or chronic [26]. Acute pain is a sharp pain of neuropathic origin characterized by burning, shooting, stinging, and stabbing. It results from the fast evolution of the inflammation of cysts, nodules, and abscesses. On the other hand, a more advanced inflammatory HS stage may cause chronic pain, which is a stimulus-dependent, nociceptive pain characterized by gnawing, aching, tenderness, and throbbing. In HS with a chronic course, nociceptive pain may interact with neuropathic pain onset acutely, resulting in mixed pain [6,27]. Scheinfeld [28] recognized seven types of HS associated pain: “(1) neuropathic pain; (2) inflammatory/joint-related pain; (3) non-inflammatory/non-neuropathic pain; (4) ischemic pain; (5) pain related to inflammatory mediators, such as tumor necrosis factor alpha (TNFa) and IL-1β; (6) pain associated with depression, anxiety and emotional stress; and (7) pain associated with arthritis.” Management of pain in HS is complex. Neuropathic pain has to be treated by anti-inflammatories [18], while nociceptive pain needs antidepressants or anticonvulsants [29], so a combined treatment may be beneficial for patients.

Pain is reported by a large majority of patients with HS. For example, in the study by Matusiak et al. [19], 77.5% of patients reported having pain in the previous week, similarly to our study, where the prevalence of patients who scored 1 or more at the VAS pain was 85%.

A more recent and extensive study of 1795 HS patients [30] showed that 83.6% of them reported HS-related pain in the previous 24 h. Pain was associated with clinical severity and with multiple affected skin areas. A strong positive correlation was found between pain assessed by numerical scale rating and quality-of-life impairment [31]. Pain occurs at each relapse of the disease in addition to foul-smelling discharge. Moreover, diagnostic delay, which still characterizes HS [32], prevents many patients from receiving adequate treatment in time, thus decreasing symptoms and improving quality of life. HS is considered the most distressing and burdensome dermatological condition due to its quality-of-life impairment. In fact, DLQI scores of HS patients are markedly higher than those of patients affected by other chronic dermatoses, such as psoriasis, chronic urticaria, atopic dermatitis, acne, or alopecia [12]. Von der Werth and Jemec [10] showed that, among patients with HS, the item of the Dermatology Life Quality Index (DLQI) with the highest score was the one concerning pain, itch, soreness, and stinging. In a qualitative study [9], patients reported various emotional reactions to HS, such as irritation, anger, and sadness, due to the fact that the lesions looked ugly, smelled, appeared quite frequently, and were painful.

Pain is thus a crucial component of quality-of-life impairment in HS even more than disease severity [19]. It has several consequences on patients’ daily life, such as sleep. In fact, pain is the major responsible factor for insomnia among HS sufferers, affecting subjective sleep quality, sleep duration, and daytime dysfunction [33]. Pain may also be a reason for absenteeism from work, with an impact on patients’ professional careers [12].

In research and particularly in clinical trials, it is important to measure outcomes, such as pain, to evaluate the efficacy of a treatment or to study the natural history of the disease. Some studies on pain in HS have used specific instruments [29]. The advantage of such instruments is that they allow to describe in detail the type and the intensity of pain and thus to define its peculiar characteristics in a disease. In the present study, we used three different methods to measure pain, each evaluating it from a different perspective. In fact, the question of the Skindex-17 refers to the frequency of pain during the previous week, the two questions of the BP scale of the SF-36 ask for pain intensity and the extent to which pain interfered with daily activities in the four previous weeks, and the VAS evaluates the intensity of pain during the previous week. When the aim is to evaluate the intensity of pain in everyday clinical practice, a simple measure can be adequate, such as a VAS for pain [34]. It provides a numerical definition of the intensity of pain, which can be easily used to compare pain over time and between patients. On the other hand, the two questions from the Skindex-17 and the SF-36 give more detailed information investigating also the frequency of pain and the impact on daily life. However, they have to be extrapolated from questionnaires, and thus they are more useful in clinical research than in practice. In our study, the three scores gave similar results when comparing health-related quality of life measured using the SF-36 in patients with different levels of pain. In particular, quality of life was always significantly lower in patients with high levels of pain when using all three measurements compared to patients with a low level for all physical and mental components of the SF-36. Differences were also significant between levels of the pain scores in pairs except for VAS pain, where differences were observed in the physical scales PF and BP for all comparisons, in RP between the scores 5–6.9 vs. ≥7, and in SF and RE between categories <5 vs. 5–6.9. This may be due to the peculiar characteristic of the VAS measurement but also to the arbitrary categorization of the VAS score, which was based on the distribution of the scores. The cross-sectional design of the present study does not allow to define the direction of the association between pain and psychological problems. It is very likely that the association is bidirectional. In fact, it has been shown that patients with HS with psychiatric comorbidities had more intense pain [6]; however, pain may also cause or exacerbate depression or anxiety.

The strong association observed in the present study between pain and clinical severity corroborates our hypothesis that pain may be a good proxy for the evaluation of clinical severity in HS. Information on pain may be easily obtained by a simple measurement, such as a VAS scale, or a question extrapolated from a questionnaire. The fact that the patient may be considered as an expert in the evaluation of her/his own disease severity has been confirmed by research data. It has been shown that patient self-assessments of flare activity and pain are strongly associated with morphological changes identified using ultrasound [35]. Therefore, the assessment of pain by the patient might also be a strong indicator of the degree of present inflammation in HS.

In conclusion, pain is a crucial hallmark of HS and an essential factor in defining health status of patients. It should be a privileged therapeutic target in patients suffering from this chronic and sometimes disabling dermatosis. Moreover, changes in pain severity may be a good proxy measure of the efficacy of a treatment.

## Figures and Tables

**Figure 1 jcm-10-03648-f001:**
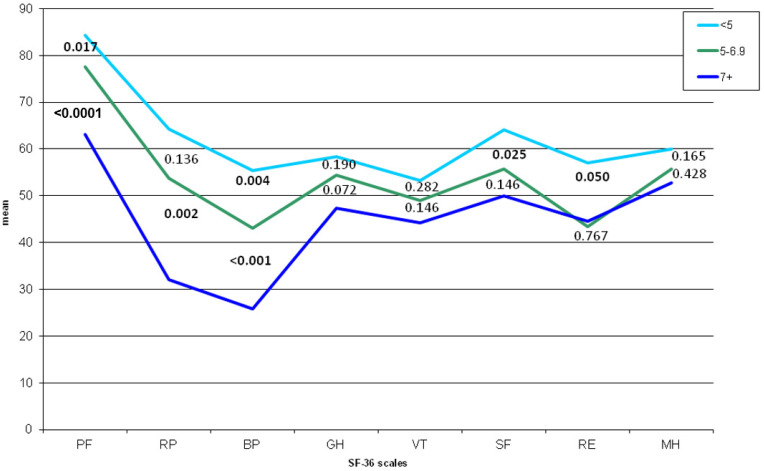
Mean values of the SF-36 scale scores in patients with different levels of VAS pain scores. PF, physical functioning; RP, role—physical; BP, bodily pain; GH, general health; VT, vitality; SF, social functioning; RE, role—emotional; MH, mental health. Numbers indicate *p*-values from Mann–Whitney U test between the mean scores of the 8 scales. Comparison between VAS < 5 and VAS ≥: PF: <0.001; RP: <0.001; BP: <0.001; GH: <0.001; VT: 0.002; SF: <0.001; RE: 0.026; MH: 0.019.

**Figure 2 jcm-10-03648-f002:**
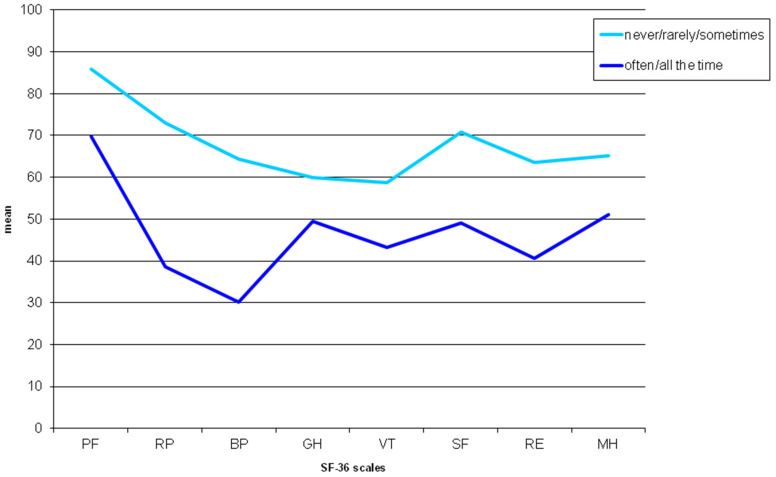
Mean values of the SF-36 scale scores in patients answering “never” or “rarely/sometimes” vs. patients answering “often/always” to question 1 of the Skindex-17 (“My skin hurts”). PF, physical functioning; RP, role—physical; BP, bodily pain; GH, general health; VT, vitality; SF, social functioning; RE, role—emotional; MH, mental health. All *p*-value < 0.001 from Mann-Whitney U test.

**Figure 3 jcm-10-03648-f003:**
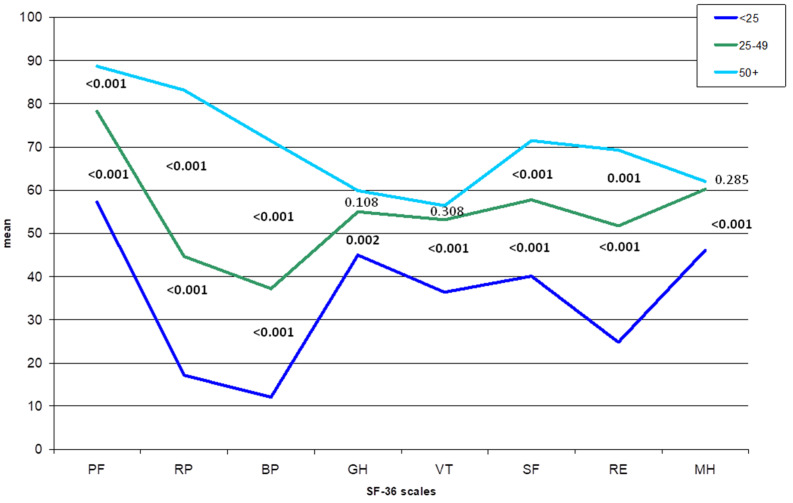
Mean values of the SF-36 scale scores in patients with different levels of Bodily Pain (BP) scores from the SF-36 questionnaire. PF, physical functioning; RP, role—physical; BP, bodily pain; GH, general health; VT, vitality; SF, social functioning; RE, role—emotional; MH, mental health. Numbers indicate *p*-values from Mann-Whitney U test between the mean scores of the 8 scales. Comparison between BP < 25 and BP = 50 or higher: all *p*-values < 0.001.

**Table 1 jcm-10-03648-t001:** Mean values of clinical severity variables according to pain in patients with hidradenitis suppurativa.

Variable	Level	N (%)		IHS4 *Mean*	Sartorius *Mean*	Hurley *Mean*	Nr Fistulae *Mean*	Nr Abscesses *Mean*	Nr Nodules *Mean*	Nr Scars *Mean*
	overall	341 (100)		19.8	58.5	1.7	2.0	2.1	7.6	10.0
**Skindex-17** **Item 1**	(a) never (b) rarely/sometimes (c) often/all the time	16 (5.1)		6.1	35.2	1.4	0.3	0.2	4.7	9.1
		96 (30.4)		13.2	46.1	1.6	1.1	1.4	5.9	8.3
		204 (64.5)		24.6	68.9	1.9	2.6	2.7	8.7	11.2
			*p **	<0.001	<0.001	0.001	0.002	0.011	0.008	0.315
			*p ***							
			a–b	0.568	0.823	0.384	0.760	0.595	0.862	0.982
			b–c	0.001	0.001	0.007	0.007	0.039	0.015	0.295
			a–c	0.017	0.053	0.013	0.082	0.083	0.160	0.861
**SF-36** **BP scale**	(d) ≥50 (e) 25–49 (f) 25	116 (39.4)		10.6	39.6	1.5	0.9	0.8	5.8	7.8
		87 (29.6)		15.3	49.4	1.7	1.6	1.3	6.4	8.3
		91 (31.0)		30.3	76.4	2.0	3.6	3.2	9.5	11.0
			*p **	<0.001	<0.001	<0.001	<0.001	<0.001	0.002	0.300
			*p ***							
			d–e	0.287	0.344	0.024	0.357	0.480	0.814	0.971
			e–f	<0.001	0.001	0.005	0.001	<0.001	0.025	0.472
			d–f	<0.001	<0.001	<0.001	<0.001	<0.001	0.002	0.299
**VAS pain**	(g) <5 (h) 5–6 (i) ≥7	150 (44.0)		12.4	42.2	1.6	1.1	1.2	5.5	7.2
		68 (19.9)		16.8	54.0	1.7	1.6	1.6	7.6	9.5
		123 (36.1)		30.5	80.9	2.0	3.4	3.4	10.2	13.5
			*p **	<0.001	<0.001	<0.001	<0.001	<0.001	<0.001	<0.001
			*p ***							
			g–h	0.435	0.278	0.441	0.734	0.765	0.174	0.560
			h–i	0.001	0.003	0.010	0.007	0.012	0.063	0.182
			g–i	<0.001	<0.001	<0.001	<0.001	<0.001	<0.001	0.002

* From one-way ANOVA; ** From Tukey’s post-hoc test. BP, Bodily Pain; IHS4, International HS Severity Score System; VAS, Visual Analog Scale.

## Data Availability

The data that support the findings of this study are available from the corresponding author upon reasonable request.

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
