# Peer review of "Pain as Defining Feature of Health Status and Prominent Therapeutic Target in Patients with Hidradenitis Suppurativa"

_jcm, 2021, doi:10.3390/jcm10163648_

Round 1

Reviewer 1 Report

In this work, Sampogna et al explore the relationship between pain and disease severity in HS patients. They concluded that HS severity is associated with pain and that the number of inflammatory lesions are significantly associated with the three severity measures of pain.

Author Response

In this work, Sampogna et al explore the relationship between pain and disease severity in HS patients. They concluded that HS severity is associated with pain and that the number of inflammatory lesions are significantly associated with the three severity measures of pain.

We thank the reviewer for her/his kind comment.

Reviewer 2 Report

Dear authors,

I have read with great interest your article about pain in HS. It is an important topic and clinicians should be encouraged to take it into consideration in treatment process.
Nevertheless, I have some minor issues with the manuscript:
1. I believe references should be divided with spaces with words before them.
2. Recently the biggest study on Pain in HS was published. I did not find it mentioned in your study and as a biggest one I believe your results should be compared to it. Here I attach the papers doi:
A. https://doi.org/10.2340/00015555-3724
B. https://doi.org/10.3390/life11010034
3. Provide EC number
4. Please improve the visibility of table/figure explanations. Now they mix with article text.
5. Please mind the punctuation e.g. et al. 

Author Response

I have read with great interest your article about pain in HS. It is an important topic and clinicians should be encouraged to take it into consideration in treatment process.

We thank the reviewer for her/his kind comment.

Nevertheless, I have some minor issues with the manuscript:

  1. I believe references should be divided with spaces with words before them.                                                                                                           We separated the reference number from the previous word, as suggested.

  1. Recently the biggest study on Pain in HS was published. I did not find it mentioned in your study and as a biggest one I believe your results should be compared to it. Here I attach the papers doi: https://doi.org/10.2340/00015555-3724 https://doi.org/10.3390/life11010034

We thank the reviewer for the suggestion. Indeed, the papers had not yet been published when we wrote the manuscript. We have added the two references and commented on the results in the Discussion.

  1. Provide EC number

We added the EC number in the Methods section.

  1. Please improve the visibility of table/figure explanations. Now they mix with article text.

We tried to better separate tables and figures from the text.

  1. Please mind the punctuation e.g. et al.

We corrected as suggested.

Reviewer 3 Report

The authors showed the relationship between severity of HS and pain. These data are important for Dermatologists involved in the treatment of HS. I think however that there are some improvements that should be made before publication.

1) The authors use three methods to measure pain, but the differences between the three methods are not clear; the advantages and disadvantages of the three methods should be described.

2)Tukey's test should be used for statistics between the three groups.

3) The authors stated that clinical symptoms such as fistulae and abscesses correlate with pain scores. This is a very important point and a table should be created to describe it in detail.

4) Correlation between each item of SF-36 and pain is described, but not mentioned in discussion. 

Author Response

The authors showed the relationship between severity of HS and pain. These data are important for Dermatologists involved in the treatment of HS.

We thank the reviewer for her/his comment.

I think however that there are some improvements that should be made before publication.

1) The authors use three methods to measure pain, but the differences between the three methods are not clear; the advantages and disadvantages of the three methods should be described.

We added a sentence in the description of the BP in the Methods section, and a comment in the Discussion: “In the present study, we used three different methods to measure pain, each evaluating it from a different perspective. In fact, the question of the Skindex-17 refers to the frequency of pain during the previous week, the two questions of the BP scale of the SF-36 ask for pain intensity and the extent to which pain interfered with daily activities in the 4 previous weeks, and the VAS evaluates the intensity of pain during the previous week. When the aim is to evaluate the intensity of pain in every-day clinical practice, a simple measure can be adequate, such as a VAS for pain [37]. It provides a numerical definition of the intensity of pain, which can be easily used to compare pain overtime and between patients. On the other side, the two questions from the Skindex-17 and the SF-36 give more detailed information, investigating also the frequency of pain and the impact on daily life. However, they have to be extrapolated from questionnaires, and thus they are more useful in clinical research than in practice.”

2)Tukey's test should be used for statistics between the three groups.

We added the results of the Tukey’s post-hoc test in Table 1.

3) The authors stated that clinical symptoms such as fistulae and abscesses correlate with pain scores. This is a very important point and a table should be created to describe it in detail.

The association between clinical symptoms and pain scores is shown in Table 1. Mean numbers of fistulae, abscesses, nodules and scars are compared in different levels of pain scores. We apologize if we did not understand the comment, however it does not seem necessary to add other analyses.

4) Correlation between each item of SF-36 and pain is described, but not mentioned in discussion.

We had already mentioned such information in the Discussion. We gave more details writing: “The three scores gave similar results when comparing health-related quality of life measured using the SF-36 in patients with different levels of pain. In particular, quality of life was always significantly lower in patients with high level of pain, using all three measurements, compared to patients with a low level, for all physical and mental components of the SF-36. Differences were also significant between levels of the pain scores in pairs, except for VAS pain, where differences were observed in the physical scales PF and BP for all comparisons, in RP between the scores 5-6.9 vs ≥7, and in SF and RE between categories <5 vs 5-6.9. This may be due to the peculiar characteristic of the VAS measurement, but also to the arbitrary categorization of the VAS score, that was based on the distribution of the scores.”